# Single-Molecule Super-Resolution Imaging of T-Cell Plasma Membrane CD4 Redistribution upon HIV-1 Binding

**DOI:** 10.3390/v13010142

**Published:** 2021-01-19

**Authors:** Yue Yuan, Caron A. Jacobs, Isabel Llorente Garcia, Pedro M. Pereira, Scott P. Lawrence, Romain F. Laine, Mark Marsh, Ricardo Henriques

**Affiliations:** 1MRC Laboratory for Molecular Cell Biology, University College London, London WC1E 6BT, UK; yue.yuan.17@ucl.ac.uk (Y.Y.); caron.a.jacobs@gmail.com (C.A.J.); pmatos@itqb.unl.pt (P.M.P.); scott.lawrence@ucl.ac.uk (S.P.L.); 2SAMRC/NHLS/UCT Molecular Mycobacteriology Research Unit, Department of Pathology, Institute of Infectious Disease and Molecular Medicine, University of Cape Town, Cape Town 7925, South Africa; 3Wellcome Centre for Infectious Diseases Research in Africa, University of Cape Town, Cape Town 7925, South Africa; 4Department of Physics and Astronomy, University College London, London WC1E 6BT, UK; i.llorente-garcia@ucl.ac.uk; 5Bacterial Cell Biology, MOSTMICRO, Instituto de Tecnologia Química e Biológica António Xavier, Universidade Nova de Lisboa, 2780-157 Oeiras, Portugal; 6The Francis Crick Institute, London NW1 1AT, UK; 7Instituto Gulbenkian de Ciência, 2780-156 Oeiras, Portugal

**Keywords:** HIV-1 entry, viral receptor, nanoscale cluster, super-resolution microscopy, CD4, STORM, quantitative analysis, modelling

## Abstract

The first step of cellular entry for the human immunodeficiency virus type-1 (HIV-1) occurs through the binding of its envelope protein (Env) with the plasma membrane receptor CD4 and co-receptor CCR5 or CXCR4 on susceptible cells, primarily CD4^+^ T cells and macrophages. Although there is considerable knowledge of the molecular interactions between Env and host cell receptors that lead to successful fusion, the precise way in which HIV-1 receptors redistribute to sites of virus binding at the nanoscale remains unknown. Here, we quantitatively examine changes in the nanoscale organisation of CD4 on the surface of CD4^+^ T cells following HIV-1 binding. Using single-molecule super-resolution imaging, we show that CD4 molecules are distributed mostly as either individual molecules or small clusters of up to 4 molecules. Following virus binding, we observe a local 3-to-10-fold increase in cluster diameter and molecule number for virus-associated CD4 clusters. Moreover, a similar but smaller magnitude reorganisation of CD4 was also observed with recombinant gp120. For one of the first times, our results quantify the nanoscale CD4 reorganisation triggered by HIV-1 on host CD4^+^ T cells. Our quantitative approach provides a robust methodology for characterising the nanoscale organisation of plasma membrane receptors in general with the potential to link spatial organisation to function.

## 1. Introduction

Cell surface receptor binding is a key step in cell infection by viruses, initiating processes that allow viral particles to cross the plasma membrane and deliver their genetic material to the host cell [1]. To infect CD4^+^ T cells, and other immune cells such as macrophages, the human immunodeficiency virus type-1 (hereafter referred to as HIV) requires binding to the surface glycoprotein CD4, and either CCR5 or CXCR4 (depending on strain tropism) as co-receptors. Biochemical and structural studies have led to a well-developed biomechanical model of the conformational changes in the viral envelope glycoprotein (Env) triggered by receptor/co-receptor binding, that lead to the formation of a fusion pore between the plasma membrane and the viral envelope. However, understanding of how receptor molecules are recruited to cell surface-bound virus particles is currently limited [2].

Previous work has suggested that receptor clustering is crucial for many receptor-ligand signalling interactions [3]. For example, on T cells, T-cell receptors (TCRs) have been shown to coalesce into nanoclusters within and around immune synapses before signal transduction [4,5,6]. Viral protein and cell-surface receptor organisation may also play a role in the local recruitment of molecules needed for successful HIV entry [7,8,9]. Because the Env subunit gp120-CD4 single inter-molecular bonds are short-lived (approximately 0.24 s lifetime) [10,11] compared to the typical duration of virus entry (in the order of minutes from receptor binding to fusion), multiple Env-CD4 interactions and CD4 receptor clustering are likely to be required for HIV entry [2]. In terms of the virus itself, previous studies have established that Env trimers redistribute and cluster on the surface of virions following Gag cleavage and maturation [12]. Given that the number of Env trimers per HIV virion is low (approximately 10 [13,14,15]), this clustering facilitates the formation of multiple receptor interactions for virus entry. Although the number of Env trimers required for virus entry is currently controversial [2], there may be a minimal local requirement for Env proteins on HIV particles and receptors on the target cell membrane for successful virus binding and fusion.

Little is known about how HIV receptor organisation on the target cell surface may facilitate, or be modulated by, virus binding. To date, most of the evidence for redistribution and clustering of cell-surface CD4 and co-receptors to sites of virus binding comes from confocal immunofluorescence microscopy studies with spatial resolution limited to approximately 200–300 nm (more than 20 times the size of individual receptor molecules) [16,17,18,19,20,21,22]. Super-resolution microscopy can study cellular organisation at the scale of an infecting viral particle (approximately 120 nm) [23]. In particular, single-molecule localisation microscopy (SMLM) techniques such as stochastic optical reconstruction microscopy (STORM) [24] allow individual cell-surface proteins to be mapped on intact cells [5,25,26,27]. Moreover, coordinate maps of all detected localisations can be analysed using point pattern methods, such as DBSCAN (density-based spatial clustering of applications with noise) [28], allowing the quantification of single-molecular nanoscale information. SMLM not only allows us to understand single molecule assemblies, but also provides information on potential interaction by protein colocalisation analysis, such as coordinate-based colocalisation analysis (CBC) [29]. Clus-DoC, an SMLM data analysis platform, which unifies DBSCAN and CBC for cluster detection and colocalisation, facilitates implementation of these analytical approaches and has previously been used to examine T-cell signalling by characterising the geometric relationships between phosphorylated and non-phosphorylated TCR [30].

In addition to distribution analysis, SMLM data can be used to count the number of fluorescently labelled molecules in a dataset, as demonstrated by estimations of the number of CD4 molecules per cluster in untreated T cells [4], TCRs upon cell activation [27], and glutamate receptors in presynaptic zones [31]. It has also recently been used to estimate the number of Influenza A virus (IAV) receptors, and to characterise their reorganisation during virus binding [32]. In the latter study, the authors established a pipeline to count the approximate number of molecules per cluster from STORM data and discovered that co-clustering between receptor sialic acid and epidermal growth factor receptor serves as an initial platform for IAV binding and signalling [32]. 

Although three recent super-resolution microscopy studies have reported CD4 cluster sizes in the range approximately 100–350 nm on resting T cells [4,33,34], here we used STORM imaging and a robust analytical pipeline to characterise the membrane distribution of CD4 at the single-molecule, nanoscale level and show that HIV binding induces rapid localised clustering of CD4 on CD4^+^ T cells. Using this imaging-based approach, we measured a 3-to-10-fold increase in the diameter of HIV-associated CD4 clusters, and an increased number of CD4 molecules per cluster (13–16 molecules, compared to 1–4 molecules in the absence of HIV). We further explored the reorganisation of CD4 clusters with statistical modelling: Cluster-size specific Poisson statistics were used to model the distributions of CD4 molecules. We found that the receptor-number distribution in HIV-bound cells differs from that predicted by a random distribution of receptors. Together, these approaches provide a novel view of the initial events of HIV binding and entry and indicate a link between a functional role and the spatial organisation of cell-surface receptors. The imaging and analytical pipeline we set out here can be used to explore the nanoscale distribution and function of other cell-surface receptors, opening the way to new insights into the molecular events underlying signal transduction. 

## 2. Materials and Methods

### 2.1. Cell Culture

SupT1 ([VB] ATCC^®^ CRL-1942™) is a CD4^+^/CCR5^−^ cell line derived from a T-cell lymphoblastic lymphoma. SupT1-R5 is a stable CCR5^+^ derivative of SupT1 provided by James A. Hoxie (University of Pennsylvania). Suspension cultures of SupT1 and SupT1-R5 were maintained in phenol red-free RPMI 1640 medium (Life Technologies, 32404-014) supplemented with 10% foetal bovine serum (FBS; Sigma-Aldrich, Dorset, UK, F9665), GlutaMAX Supplement (Life Technologies, Paisley, UK, 35050-038), 50 U/mL penicillin, and 50 μg/mL streptomycin at a density of 1 × 10^5^–1 × 10^6^ cells/mL at 37 °C with 5% CO_2_.

Human embryonic kidney (HEK) 293T cells (ATCC^®^ CRL-3216™) were cultured in Dulbecco’s Modified Eagle Medium (DMEM; Life Technologies, Paisley, UK, 31053-028) supplemented with 10% FBS, GlutaMAX Supplement, 50 U/mL penicillin, and 50 μg/mL streptomycin at 37 °C with 5% CO_2_. 

HeLa-TZM-bl cells were cultured in DMEM, supplemented with 10% FBS and 1% GlutaMAX Supplement at 37 °C with 5% CO_2_. 

### 2.2. Antibody Conjugation

The monoclonal antibody OKT4 (anti-CD4) [35] was conjugated with Alexa Fluor fluorophores using NHS-ester chemistry Lightning Link Kits (Innova Biosciences, Cambridge, UK). Briefly, 10 μg IgG (1 mg/mL in PBS) were mixed with 1 μL of LL-Modifier reagent, before the addition of 2–3 molar equivalents of NHS ester-functionalised Alexa Fluor 568 or Alexa Fluor 647 (from 10 mg/mL stocks in DMSO). The reaction mixture was incubated at room temperature (RT, 23 °C) for 3–4 h. Subsequently, 1 μL of LL-Quencher was added to terminate the reaction. Unreacted fluorophore was removed by diluting the reaction volume to 500 μL and centrifugation through 3 kDa MWCO Amicon Ultra centrifugal filter columns at 14,000 relative centrifugal force (RCF) for 15 min. This washing step was repeated three times. Protein concentration and protein:dye ratio were determined by spectrophotometric measurement of absorbance at 280 nm and 280 nm vs. 647 nm or 568 nm, respectively. Antibody cell labelling specificity was confirmed by comparison of immunolabeling signals of CD4^+^ (Jurkat and SupT1) and CD4^−^ (HEK293T) cells by epifluorescence microscopy. 

### 2.3. Phorbol Ester Stimulation

SupT1-R5 cells (1 × 10^4^ cells per sample) were pelleted and resuspended in 20 μL cold RPMI-1640 with 0.4% FBS and 6 μg/mL OKT4-Alexa Fluor 647. Cells were incubated on ice for 60 min before washing, which was carried out by making up the cell volume to 10 mL in cold RPMI with 5% FBS and centrifuging the cells at 300 RCF for 6 min at 4 °C. The supernatants were carefully aspirated and the cells washed twice in 10 mL cold RPMI-1640 with 5% FBS, as described above. Cells were resuspended in 60 μL of cold RPMI-1640, with 5% FBS, and allowed to settle on Poly-L-lysine (PLL)-coated coverslips (100 μg/mL in ddH_2_O, Sigma-Aldrich, Dorset, UK, P8920) at 4 °C for 40 min. Treated samples were transferred to 37 °C RPMI-1640 with 5% FBS, with or without 2 mg/mL Phorbol-12-myristate-13-acetate (PMA) or 4-α-Phorbol-12-myristate-13-acetate (4α-PMA) and incubated for 15 min at 37 °C. Thereafter, cells were returned to cold RPMI-1640 for 1–5 min and fixed with cold 4% paraformaldehyde (PFA) for 10 min. Control samples were transferred directly to cold 4% PFA without warming. Samples in PFA were then warmed to 37 °C over 20 min, washed five times in PBS, and stored in PBS until mounting for imaging.

### 2.4. Virus Preparation

HEK293T cells were seeded at 2.25 × 10^6^ cells per T75 culture flask to give <50% cell confluency the following day. 1.5 mL of OptiMEM was mixed with 15 μg of HIV_JR-CSF_ proviral DNA and incubated at RT for 5 min, before gently mixing with 45 μL Fugene 6 equilibrated to RT. Subsequently, cell medium was replaced with 15 mL antibiotic-free DMEM containing 10% FBS, after which the transfection reaction mixture was added and the cells were incubated at 37 °C with 5% CO_2_. After 48 h, the culture medium was collected and centrifuged at 500 RCF for 10 min. The supernatant was transferred to Beckman ultracentrifuge tubes underlaid with a 5 mL cushion of sterile 20% sucrose in PBS. The tubes were topped up with complete media and the virus pelleted through the sucrose cushion by ultracentrifugation at 98,000 RCF for 2 h at 4 °C. The supernatant was carefully aspirated to preserve the viral pellet, which was then resuspended in DMEM, aliquoted, and stored at −80 °C in liquid nitrogen.

### 2.5. Virus Titration

HeLa-TZM-bl cells were seeded at 1 × 10^3^ cells per well in a 96 well microtiter plate. After 12–18 h, the cells were infected with HIV in a series of two-fold dilutions from 1 in 2 to 1 in 256 in a final volume of 400 μL. At 6 h post-infection (hpi), 8 μg Q4120 was added to each well to prevent syncytia formation. Cells were washed in PBS at 36 hpi and fixed in 4% PFA for 30 min at RT, followed by incubation in 0.1% PFA overnight at 4 °C. Cells were then washed three times with PBS at RT, quenched in 50 mM NH_4_Cl for 10 min, blocked and permeabilised with PBS supplemented with 1% FBS and 0.1% Triton for 15 min at RT, and finally washed with 1% FBS in PBS. Cells were incubated with rabbit antiserum to HIV p24/p55 Gag (ARP432; NIBSC Centre for AIDS Reagents, South Mimms, UK) at 1:500 in PBS containing 1% FBS for 1 h at RT, washed three times with PBS/1% FBS and incubated with Goat anti-rabbit IgG-Alexa Fluor 488 (H+L; Life Technologies, A-11008) and DAPI for 20–30 min. Cells were washed with PBS before imaging the plates using a Perkin Elmer Opera Phenix high-throughput plate reader with a 20× air objective. Images (9 per well) were analysed using Columbus Image Analysis software.

### 2.6. HIV Binding

1 × 10^4^ SupT1 and SupT1-R5 were pre-incubated with OKT4-Alexa Fluor 647 (6 μg/mL) in a total volume of 20 μL for 15 min at RT. The cells were then cooled to 4 °C. Initially three different HIV multiplicities of infection (MOI) were tested to find a balance between having a sufficient number of bound particles per field of view (FOV) and a relevant number of viruses. MOIs of 15, 30 and 50 infectious units per cell were tested. We observed an average of 1, 23 and 42 HIV particles per cell, respectively. In order to maximise the experimental throughput while remaining within reasonable bounds of viral infection, we chose an MOI of 30 as the being optimal. After 1 h, free virus was removed by diluting the cells in 10 mL cold RPMI-1640 and centrifugation at 500 rpm for 10 min. For 0 min warm up samples, the cell pellets were resuspended in 300 μL cold serum-free RPMI-1640 and allowed to settle onto PLL-coated μ-Slide 8 Well dishes (Ibidi GmbH, Gräfelfing, Germany) for 40 min. For 1 min warm up samples, the cells were resuspended in 50 μL serum-free RPMI-1640 and transferred to a water bath at 37 °C. After 1 min, 10 mL cold serum-free RPMI-1640 were added to the sample tube, followed by centrifugation at 500 rpm for 10 min. The cell pellets were then resuspended in 300 μL serum-free RPMI-1640 and allowed to settle onto PLL-coated μ-Slide 8 Well dishes for 40 min. Settled cells were fixed directly by incubation with 4% PFA in PBS for 30 min at RT (control sample), or the cold media was replaced with pre-warmed media, and samples incubated at 37 °C for 1 min before fixing as described above. After fixation, samples were washed three times with PBS and permeabilised in 0.1% Tween-20 in PBS for 5 min, blocked for 20 min in PBS containing 4% BSA, and labelled with anti-HIV Gag antiserum as above. The cells were then washed three times in PBS and incubated with 1:500 Goat-anti-rabbit-Alexa Fluor 568 (Life Technologies, A-11036) for 1 h at RT, then washed five times in PBS (5 min for each wash), subjected to a second round of fixation with 4% PFA in PBS for 30 min at RT, washed five times in PBS and stored in PBS before mounting for analysis by microscopy.

### 2.7. gp120 Binding

SupT1 and SupT1-R5 cells (1 × 10^4^ cells per sample) were pelleted and resuspended in 20 μL cold RPMI-1640 with 0.4% FBS and 6 μg/mL OKT4-Alexa Fluor 647, with or without 1 μg/mL HIV_BaL_ gp120 (NIH AIDS Reagents Program). Cells were incubated on ice for 60 min before being diluted in 10 mL cold serum-free RPMI-1640 and centrifuging at 300 RCF for 6 min at 4 °C. The supernatants were carefully aspirated, and the cells resuspended and washed twice in 10 mL cold serum-free RPMI-1640. The cells were then resuspended in 60 μL cold serum-free RPMI-1640 and allowed to settle on PLL-coated coverslips (100 μg/mL in ddH_2_O, Sigma-Aldrich, Dorset, UK, P8920), on ice, for 40 min. Control samples were transferred directly to cold 4% PFA. Treated samples were transferred to 37 °C serum-free RPMI-1640 for 1 min, before returning to 1 mL cold serum-free RPMI-1640 for 5 min to rapidly cease cells’ reaction to treatment. The cells were then fixed with cold 4% PFA for 10 min, before warming to 37 °C over 20 min. Samples were washed five times in PBS and stored in PBS until mounting for imaging.

### 2.8. Imaging

Samples prepared on coverslips were mounted on parafilm-formed gaskets (as described in [36]) with STORM buffer (150 mM Tris pH 8, 1% glycerol, 1% glucose, 10 mM NaCl, 1% beta-mercaptoethanol, 0.5 mg/mL glucose oxidase, and 40 μg/mL catalase) and sealed with clear nail varnish. For samples on Ibidi GmbH slides, the sample wells were filled with STORM buffer, and the lid sealed with High-Performance Black Masking Tape (Thorlabs, UK).

Imaging was carried out on a Zeiss Elyra PS.1, with an alpha Plan-Apochromat DIC M27 Elyra 100 × 1.46 Numerical Aperture (NA) oil objective, additional 1.6× optovar magnification, and Andor iXon 897 electron multiplication CCD (EMCCD) camera, yielding a pixel size of 100 nm. STORM datasets of 15,000 sequential frames were acquired in Total internal reflection fluorescence (TIRF) configuration, using 33 ms exposure time, with 642 nm or 561 nm excitation at maximum power output (approximately 3.98 kW/cm^2^ and 3.71 kW/cm^2^ on the sample, respectively). Fluorophore photoswitching was dynamically controlled using periodic 405 nm illumination at the intensity of approximately 0–0.0586 kW/cm^2^ on sample laser power. EM camera gain of 300 was used. Microscope autofocus was used throughout all acquisitions.

### 2.9. Localisation Algorithm

SMLM imaging datasets were processed using the ThunderSTORM analysis plugin [37]. Initial particle localisation was performed using the local maximum method, followed by sub-pixel localisation using the integrated Gaussian model and fitting by Maximal Likelihood Estimation. Drift correction was performed post-localisation by cross-correlation (number of bins was 5.0). Reconstructed images were rendered using a normalised 20 nm Gaussian.

### 2.10. Cluster Analysis

For cluster analysis, we used Clus-DoC [38]. 30 nm Epsilon and 1 Minimum point parameters were used throughout the study. We manually selected our regions of interests (ROIs) to include as much of the plasma membrane as possible while avoiding cell-edge effects. For each cluster, we obtained the area and corresponding mean diameter, the cluster density per ROI (number of clusters detected/ROI area [μm^2^]) as well as the number of localisations for each cluster.

### 2.11. Cluster Manual Annotation

To validate the quantitative results from Clus-DoC [38], we manually annotated clusters in Fiji [39]; cross-sections of identified clusters were drawn individually. The intensity profile was plotted and fitted with a Gaussian distribution of standard deviation σ. The cluster diameter was estimated using the Full-Width Half Maximum (FWHM) as d = 2.35 σ. 

### 2.12. Channel Registration

Channel registration was performed using a chromatic aberration correction plugin developed by the Jalink lab (https://jalink-lab.github.io/). Fiducial bead-coated coverslips were imaged for each experiment under both 647 nm and 568 nm channels. Fiducial bead images were used as references for estimating the transformation between channels. This correction was directly measured and applied to localisation data. 

### 2.13. Colocalisation Analysis

To determine the colocalisation threshold, we used post-channel registration fiducial bead images. When the optimised threshold was set to 0.4, the majority of events co-localised (>99%) with a peak of the DoC distribution at 1, indicating high colocalisation. Aside from the DoC score, we also extracted information on the percentage of colocalisation between channels and the mean cluster diameter in the colocalised and non-colocalised areas.

### 2.14. Molecular Counting

We performed molecular counting as described [32]. To calibrate the grouping parameters, we performed the same STORM imaging procedure as described in 2.9 on isolated fluorophore-conjugated antibodies. The imaging dishes were incubated with 1 μg/mL dye-conjugated anti-CD4 antibody for 15 min. The dishes were washed once and then imaged under the same experimental conditions as for the main dataset using T cells. Images were reconstructed as described in 2.10. Localisations within 30 nm were merged as one localisation frame by frame to form a new coordinates map and temporally binned to extract calibration parameters for molecular counting. All localisation processing was performed using custom-written MATLAB (MathWorks) scripts (kindly provided by Christian Sieben). Alignment of localisations from individual molecules also allowed for estimation of the localization precision, as described in [32].

### 2.15. Theoretical Poisson Statistical Model

We used a theoretical model based on cluster-size specific Poisson probability distributions to determine whether our results for CD4 molecule numbers in clusters are consistent with a random distribution of receptors on the cell surface. 

We generated theoretical distributions of the number of molecules per cluster expected for a random distribution of receptors using an average CD4 surface density *n* = 200 molecules/μm^2^. This comes from considering a CD4 density of approximetaly 100,000 molecules/cell in SupT1-R5 cells [40,41] and typical T-cell shapes with a radius of approximately 10 μm. Cells were approximated as a flat disk with total surface area 2 × π*r*^2^ = 630 µm^2^. 

The expected probability of observing a number of receptors *k* in a given area *A* of the cell surface is given by the discrete Poisson probability distribution, *P*(*k*; *λ*), where *k* ≥ 0 and *λ = nA* is the mean receptor count expected when counting receptors on a patch of area *A* (*λ* is also the mean of the distribution). This simple model assumes independence of receptor-counting events and, hence, no interactions between receptors, signalling, or active processes or receptor mobility are considered. Thus, in this model, counting of multiple CD4 receptors in a given area is considered to be purely due to chance, i.e., to random statistical fluctuations.

Our imaged CD4 clusters had radii (*r_i_*) in the range approximately 10–100 nm (considering their equivalent circular areas). Their corresponding mean parameters (*λ_i_*) are therefore in the range 0.1–6. The Poisson distribution has significantly different shapes for these different values of its mean parameter, and is increasingly asymmetrical for decreasing values of *λ_i_* below 5. For the larger cluster sizes (larger *λ_i_*), the distribution mean (and peak position) shifts to higher values of *k* and the probability of counting a larger number of molecules per cluster increases. For these reasons, our model considered the different sizes of all the measured clusters to calculate the overall probability distribution of numbers of molecules per cluster that would be observed when counting receptor numbers in circular cluster areas equivalent to those occupied in our measured CD4 clusters.

The expected overall distribution of numbers of molecules per cluster is therefore the average of the Poisson distributions corresponding to all our observed cluster sizes (areas of radius *r*_*i*_). The overall probability (*p*) of counting *k* receptors is:$$P\left(k\right)=\frac{1}{N}\sum_{i=1}^{N}P\left(k;\lambda_{i}\right)$$
where we summed over all the different clusters (*i* is the cluster index and *N* is the total number of clusters measured), *P*(*k*;λ_*i*_) is the Poisson distribution for a given cluster *i* with radius *r*_*i*_ (that occupies a surface area *A*_*i*_ = π*r*^2^_*i*_), and λ_*i*_ = *n**A*_*i*_ = π*r*_*i*_^2^*n* is the corresponding mean value of the Poisson distribution for cluster *i*. 

In order to compare expected and measured distributions, we re-normalised our expected distributions by excluding and dividing by the sum of the remaining counts. This is because areas with zero receptors were not measured in our experiments. Calculations were performed using custom-written Python scripts. 

### 2.16. Statistical Analysis

All data are presented as means ± standard deviation (SD) from three independent experiments. *N* is indicated in each figure separately. Student t-tests were performed by GraphPad Prism 8 (Prism Software). Significance was calculated using unpaired Student’s *t* tests.

## 3. Results

### 3.1. Analysis of CD4 Nanoclusters on CD4^+^ T Cells by Quantitative Super-Resolution Imaging

In order to study the effect of HIV-binding on the organisation of CD4 molecules at the cell surface, we first established an experimental and analytical framework to quantitatively characterise nanoscale receptor organisation (Figure 1). To avoid potential artefacts caused by fixation and antibody-induced artificial crosslinking, we developed an experimental assay in which immunolabeling and receptor-virus binding were carried out at low temperature to minimise membrane fluidity and trafficking [42,43,44]. Although HIV-target cell-binding occurs at 37 °C in vivo, virus bound at 4 °C can display CD4-specific association and occupy an intermediary state in the cell entry process referred to as a temperature arrested state (TAS) [42,43,44]. A subsequent brief release of the temperature block, and rapid cooling prior to fixation, allowed us to capture receptor reorganisation at early stages following virus binding (Figure 1AI). We have previously carefully evaluated cell fixation protocols to preserve native membrane protein organisation [35]. Labelling of cell-surface receptors is challenging due to the potential for various labelling strategies to subtly alter the membrane organisation of the receptor of interest. We adopted an immunolabelling approach expected to minimise antibody-induced CD4 dimerisation, and confirmed that antibody binding did not lead to detectable crosslinking or perturbation of receptor distribution on the time scales of our assays (Appendix A).

We used TIRF-STORM to image CD4 in the cell membrane adjacent to the glass (Figure 1AII, Appendix A, estimated localisation precision σ_x,y_ approximately. 15 nm, and Appendix A), followed by Clus-Doc [38] to identify clusters and estimate their shape parameters (area, equivalent diameter, and localisation density; Figure 1AIII). The diameter results produced by Clus-Doc [38] were validated by comparison with manual cluster annotation, confirming the suitability of this approach (Appendix A). TIRF-STORM imaging and cluster characterisation in untreated SupT1-R5 cells revealed CD4 is arranged in small clusters of 64 nm ± 33 nm diameter (Figure 1B,C), with the majority (approximately 70%) composed of 1 to 4 CD4 molecules (Figure 1D and Appendix A).

To determine if our analysis pipeline could robustly identify changes in CD4 organisation, we used chemical induction to redistribute CD4 in a predictable and controllable manner. Treatment with the phorbol ester PMA stimulates clathrin-mediated endocytosis (CME) of CD4 by activation of a phosphorylation-dependent dileucine signal [45]. SupT1-R5 cells were treated with PMA for 15 min, as previously described [46], and imaged with TIRF-STORM. Quantification of CD4 localisations revealed the formation of large plasma membrane CD4 clusters (93 nm ± 36 nm diameter; Figure 2A,B). At the single cluster level, a clear pool of larger CD4 clusters of up to 800 nm diameter was seen (Figure 2C). Increased cluster diameters were not detected in cells fixed immediately following PMA addition, nor in those treated with non-stimulatory 4α-PMA (Figure 2B,C), both of which showed cluster diameter distributions similar to that of untreated cells (as in Figure 1C). Localisation-based estimates indicated that PMA induced an increase in the number of CD4 molecules per cluster (21–100 molecules per cluster; average of 38 ± 73) and, strikingly, a small number of clusters with as many as 500 molecules (Figure 2D). This CD4 reorganisation is an expected consequence of PMA treatment and indicates that our pipeline should detect potential HIV-induced changes in the cell-surface CD4 distribution.

### 3.2. HIV Receptor Binding Induces Changes in CD4 Clustering

Although HIV cell entry is dependent on the initial binding of CD4 molecules by the viral Env subunit gp120, how receptor binding impacts on the nanoscale organisation of CD4 is poorly understood [2]. Therefore, we set out to measure the effect of virus binding on CD4 nanoscale organisation using our quantitative pipeline. 

We incubated SupT1-R5 cells with HIV_JR-CSF_ under CD4-binding permissive (Figure 3A) or non-permissive (Figure 3B) conditions (i.e., in the presence of the neutralizing anti-CD4 antibody Q4120, which inhibits Env-CD4 binding [47]), followed by brief treatment at 37 °C before rapid cooling prior to fixation. Dual-colour TIRF-STORM imaging of immunolabeled CD4 and HIV p24 (Figure 3A,B) revealed enlarged CD4 clusters around HIV contact sites in permissive conditions, but not in non-permissive control conditions (Figure 3D,E). Using coordinate-based colocalisation analysis built into Clus-DoC, we identified and characterised CD4 clusters that were associated with cell-surface bound HIV. HIV-colocalised CD4 clusters were more than twice the diameter of non-HIV-colocalised clusters on the same cells (191 ± 6 nm vs. 71 ± 8 nm, respectively; Figure 3D). The diameters of non-associated clusters were in good agreement with those measured on untreated cells (65 ± 9 nm vs. 64 ± 33 nm, respectively). This increase in cluster diameter was not observed when HIV binding was blocked by Q4120 (65 ± 6 nm), nor in HIV-bound cells that were fixed prior to 37 °C incubation (63 ± 8 nm) (Figure 3B,E and Appendix A). In conditions permissive for receptor binding, the increase in CD4 cluster diameter was accompanied by an equivalent increase in the number of CD4 molecules per cluster; 70% of clusters consisted of 13–16 molecules, compared to 1–4 molecules per HIV-associated cluster when receptor binding was inhibited (Figure 3F). These results were further validated by HIV-CD4 colocalisation analysis which showed that around 80% of HIV particles were engaged with CD4 clusters in permissive conditions (Appendix A). In contrast, less than 5% of viral particles were associated with CD4 clusters in non-permissive conditions in the presence of Q4120 (Appendix A). This HIV-associated change in clustering was dependent on incubation at 37 °C after release of the TAS (Appendix A), and the CD4 cluster changes were abrogated by inhibition of HIV-CD4 interactions (Figure 3B,E). These data indicate that HIV-receptor binding at the plasma membrane induced an increase in local CD4 clustering around bound viral particles in a manner dependent on direct interaction between the virus and receptors. Importantly, this reorganisation did not lead to a statistically significant general increase in the density of CD4 clusters across the fields imaged (Appendix A). The increase in cluster diameters is therefore likely to be due to local accumulations of CD4 molecules (Figure 3C and Appendix A).

Next, we compared the changes in CD4 organisation induced by HIV to a theoretical Poisson statistical model used to calculate the distribution of the numbers of molecules per cluster that could be expected from a random distribution of receptors on the cell surface. An overall CD4 surface density (*n*) of 200 molecules/μm^2^ was used, based on the number of CD4 molecules per cell reported for SupT1 cells [40,41], and the different cluster sizes measured experimentally were used to generate an averaged theoretical Poisson distribution (see Figure 4A and Section 2). In untreated cells, the experimental data was similar to the modelled distribution (Figure 4B). However, in samples with HIV bound under permissive conditions, the experimentally measured data showed a clear additional peak at 13–16 molecules per cluster that cannot be explained by the theoretical random distribution model. Specifically, we measured an approximately four-fold increase in the fraction of clusters with 13–16 CD4 molecules when HIV was present (Figure 4C), with that fraction corresponding to an increase in cluster diameters (Appendix A). We also tested the effect of possible variations in cell-surface CD4 levels or cell size by generating theoretical Poisson distributions as above, using overall density values of *n* = 60 molecules/μm^2^ and *n* = 300 molecules/μm^2^. When we tested our experimental data against these models, we saw a similar clear additional peak at 13–16 molecules per cluster that was not predicted by the theoretical model (Appendix A). These results support the notion that the observed CD4 cluster reorganisation is a result of HIV binding. 

### 3.3. HIV gp120-Induced CD4 Clustering is Independent of CCR5

Our data suggest that HIV binding induces a reorganization of CD4 at the cell surface, promoting local concentrations of CD4 molecules, as indicated by an increase in CD4-cluster diameters and CD4 molecules per cluster. Based on these findings we posited that the increased CD4 clustering could be a biological requirement for productive HIV binding and/or entry. We sought to explore this hypothesis by assessing the ability of HIV Env to locally recruit CD4 using a reductionist approach. For this purpose, we analysed CD4 organisation following soluble gp120 binding, using a similar approach to that described above for HIV binding. TIRF-STORM imaging and cluster characterisation of CD4 on gp120-treated SupT1-R5 cells revealed increased cluster diameters following the release of the temperature block (84 ± 12 nm) (Figure 5A (left), Figure 5B), although to a lesser extent than that induced by intact HIV (191 ± 6 nm) (Figure 3D). This change in cluster diameters was accompanied by a higher variation in CD4 molecules per cluster—a decrease in the number of the smallest clusters of up to 4 molecules, and small increases in the number of clusters consisting of 5–8 and 13–16 molecules (Figure 5C). This indicated that upon binding, gp120 alone can locally influence CD4 distribution, although the extent of this effect was less than that induced by viral particles.

CD4 binding by gp120 in functional Env trimers stabilises an ‘open’ Env conformation that permits subsequent Env binding to a co-receptor—e.g., CCR5—which, in turn, drives membrane fusion and viral entry [43,48]. To determine whether the changes observed in cell-surface CD4 organisation following HIV binding were dependent on the presence of the co-receptor, in this case CCR5, we compared CD4 clustering on gp120-bound SupT1-R5 cells to that on SupT1 cells which do not express CCR5. No difference was detected in mean cluster diameter (86 ± 18 nm vs. 84 ± 12 nm), or in the number of CD4 molecules per cluster, in the presence or absence of CCR5 (Figure 5A (right), Figure 5B). These results suggest that local recruitment of CD4 molecules immediately following gp120 binding is not dependent on the presence of a co-receptor.

## 4. Discussions

Knowledge of the initial stages of virus infection i.e., the molecular details underpinning the interaction between viral proteins and host-cell receptors, is essential for a full understanding of the mechanism through which HIV enters cells. While HIV probably first attaches to cells through diverse interactions with various adhesion factors [49,50,51,52,53], the virus is dependent on binding to CD4 and the co-receptors CCR5 or CXCR4 for successful fusion with the target cell membrane [54]. Previous studies have pointed to a role for both Env and CD4 clustering in HIV entry [16,17,18,19,20,55,56,57], however, the relationship between the spatial distribution of receptors and the fusion process has remained largely speculative. To understand this process in more detail, we sought to characterise the nanoscale organisation of cell surface HIV receptor proteins during virus binding and entry. Here we describe the use of super-resolution SMLM and a suite of analytical tools to determine the organisation of CD4 in response to HIV binding. 

We established an experimental and analytical pipeline with the necessary range and sensitivity to detect nanoscale receptor organisation. Although SMLM provides significantly improved lateral and axial spatial resolution, most membrane receptor studies are conducted via two-dimensional imaging and neglect cell-surface nano-topography. Immobilisation of cells on a glass surface for imaging in TIRF mode can sometimes interfere with the complex cell-surface morphology. For instance, the flattening of T cells on PLL-coated glass can cause artefacts when studying membrane receptor (re)organisation [58,59,60]. In our study, we co-incubated HIV and CD4^+^ T cells in suspension before depositing the cells on PLL-coated coverslips to minimise potential artefacts induced by cell-glass interactions. 

To verify that our approach could identify nanoscale changes in CD4 organisation, we tested this pipeline using PMA, a known inducer of CD4 clustering and clathrin-mediated CD4 endocytosis [46] (Figure 1 and Figure 2). Density-based nanoscale analysis of CD4 organisation on untreated T cells revealed that CD4 is distributed into nanoscale domains containing a single molecule or small clusters of up to four CD4 molecules (Figure 1). These results agree with a growing body of evidence that cell-surface receptors are generally organised into clusters [5,25,48,61], some on a scale similar to membrane nanodomains (approximately 50–200 nm), that may correspond to the corrals organised by the cortical actin meshwork [62,63,64]. However, the term ‘cluster’ can be challenging to define in SMLM because detected localisations could represent a single molecule or small group of molecules. We adopted a definition from similar work that studied TCR distributions, specifically that nanoclusters are ‘molecules grouped at a sub-diffraction limit scale (<250 nm), such that the grouped coordinates are unlikely to be completely randomly distributed’ [65]. 

Using SMLM localisation, we were able to obtain estimates of the numbers of CD4 molecules per localisation or cluster, following the approach described by Sieben et al. [31]. Considering that the numbers of molecules counted was mostly <10 per cluster, it is important to note that these figures should be considered as rough estimates and not true numbers of molecules. The accurate determination of molecule numbers using SMLM methods remains a challenge due to labelling efficiency, re-emitting fluorophores, variable imaging conditions and differences in analytical approaches [66]. A potential undercounting in our experiments may arise from low labelling efficiency. In particular, we may underestimate the number of CD4 molecules as OKT4, the anti-CD4 antibody used in this work, has a lower binding efficiency compared to, for example, the neutralizing anti-CD4 antibody Q4120. Q4120 is more commonly used for characterising cell-surface CD4 but is unsuitable for HIV-CD4 interaction studies. In addition, we cannot completely exclude the possibility of CD4 dimerisation by the labelling antibody which, although we would expect it to be minimal, may also contribute to receptor undercounting. Although the CD4 levels per cell measured in this study (total number of molecules/the area of a region of interest) are lower than those reported previously (60 molecules/µm^2^ vs. 200 molecules/µm^2^) [40,41], given the very different measuring techniques used, the numbers are reasonably close. The post-processing calibration for molecule counting implemented here may also lead to undercounting. The process depends on the gap time distribution parameter selected when quantifying the dark time used to calibrate the experimental localisation coordinate lists. The gap time distribution fitting is a trade-off between over-merging (i.e., high confidence but underestimated protein count) and under-merging (i.e., lower confidence but higher protein count). In this study, a longer gap time was used to minimise false-positive molecular identifications. Despite these uncertainties, we can nevertheless extract useful information by emphasising the relative changes in molecule numbers under different experimental conditions. Our results consistently indicate a 3-to-10-fold increase in both cluster diameter and the number of molecules assembled around bound HIV particles (Figure 3). Interestingly, the persistence of cell-bound HIV not associated with CD4 or in assays with Q4120 can be attributed to the ability of HIV to bind to cells through interactions with adhesion factors, including integrins and lectins [49,50,51,52,53].

To further confirm that the CD4 clustering detected on cells with bound HIV was not attributable to chance, we applied statistical modelling to simulate a random distribution of receptors on the cell surface and compared our results with the experimental data (Figure 4). Based on previously reported numbers of approximately 100,000 CD4 molecules/SupT1 cell [40,41], and an average T-cell radius of 10 μm, we assumed an overall mean density (*n*) of 200 CD4 molecules/μm^2^ in our model (Figure 4). To compare this estimate with our experimental data, we estimated the number of CD4 molecules on SupT1-R5 cells labelled with Q4120 and OKT4. We measured a density of 180 molecules/μm^2^ with Q4120 and 60 molecules/μm^2^ with OKT4. Given these antibodies recognise different epitopes on the CD4 molecule, we regard these measurements as being in reasonable agreement. Nevertheless, to take possible variations in detected CD4 levels and in cell size into account, we tested our model at lower and higher mean density values of *n* = 60 molecules/μm^2^ and *n* = 300 molecules/μm^2^ (Appendix A). In all cases, our conclusions stand, i.e., HIV binding induces CD4 clustering, as opposed to clustering being a product of occasional random accumulation. This statistical model allows us to assign greater confidence to our interpretation of the data and the biological significance of receptor clustering changes. Importantly, this framework can be easily adapted to studies of other membrane-associated receptors and induced organisational changes.

Given the measured density of CD4 molecules within the HIV-induced clusters (approximately 500 molecules/μm^2^), we estimated the inter-molecular centre-to-centre distances for CD4 molecules as 40–50 nm. This distance is larger than the measured width of trimeric Env complexes of 10–15 nm [17,20,67,68]. Even considering the likely undercounting in our data, this suggests that not all gp120 molecules on an individual Env spike engage CD4. Further, given that mature HIV particles have an average of 10 Env trimers per virion [13,14,15], which can be arranged in clusters in the viral envelope [12,55]. Our measured numbers of 13–16 CD4 molecules in an HIV-associated cluster suggests an approximate ratio of 1:1 CD4 per Env trimer. Nevertheless, it remains difficult to estimate the precise Env-CD4 stoichiometry required for fusion and entry, as the organisation of the relevant Env-CD4 interactions within each virus-CD4 cluster remains to be established. For membranes to fuse, both CD4 and Env molecules, as well as other proteins, must be cleared from the zone of close approach of the opposing bilayers. Thus, CD4-Env interactions might be organised as a ring at the edge of the cell-virus interface. In part at least, this would be consistent with the observation that the measured CD4 clusters have diameters that exceed those of virus particles (typically approximately 120 nm) and point to a larger biophysical influence for Env trimer clustering than previously thought, potentially requiring flexibility in Env and/or CD4 as recently shown for the spike (S) protein of SARS-CoV-2 [69]. 

From a downstream mechanistic point of view, although signalling through the GPCR co-receptors has been identified as a process that HIV can exploit to remodel the cytoskeleton and facilitate productive infection [68], there are conflicting reports as to whether GPCR-dependent signalling through CCR5 is required for HIV entry [70,71]. The CD4 clustering we observed, coupled with the independence of gp120-induced CD4 clustering from co-receptor CCR5, suggests that HIV-induced signalling through CD4 and Lck (such as that detected by Lucera et al. [71]) can trigger CD4 clustering beyond the range of Env binding. Furthermore, CD4/Lck signalling, while not essential for HIV infection (macrophages are Lck-negative yet HIV-susceptible), may play a more significant role in facilitating efficient HIV entry in some cell types than previously appreciated. A few theoretical studies have used stochastic models that account for receptor diffusion and receptor-virus binding to predict the accumulation of HIV receptors at virus-binding sites on the surface of target cells [72,73,74]. Future comparison of model predictions and experimental data may provide further insights into the mechanisms of receptor cluster formation.

In summary, our study provides one of the first nanoscale analysis of HIV-induced spatial changes in cell-surface CD4 distribution during the initial steps of HIV infection on CD4^+^ T cells. HIV-induced CD4 spatial reorganisation might link to their functional role during HIV binding and fusion. This work contributes to our goal of better understanding how viruses spatially modulate cell-surface receptors to facilitate their entry, and how we might prevent virus infection by inhibiting not only receptor binding but also receptor spatial redistribution. The imaging and analytical pipeline we have established provides a powerful and robust toolbox to further study HIV entry as well as the properties of other virus-receptor systems and membrane proteins in general. 

## Figures and Tables

**Figure 1 viruses-13-00142-f001:**
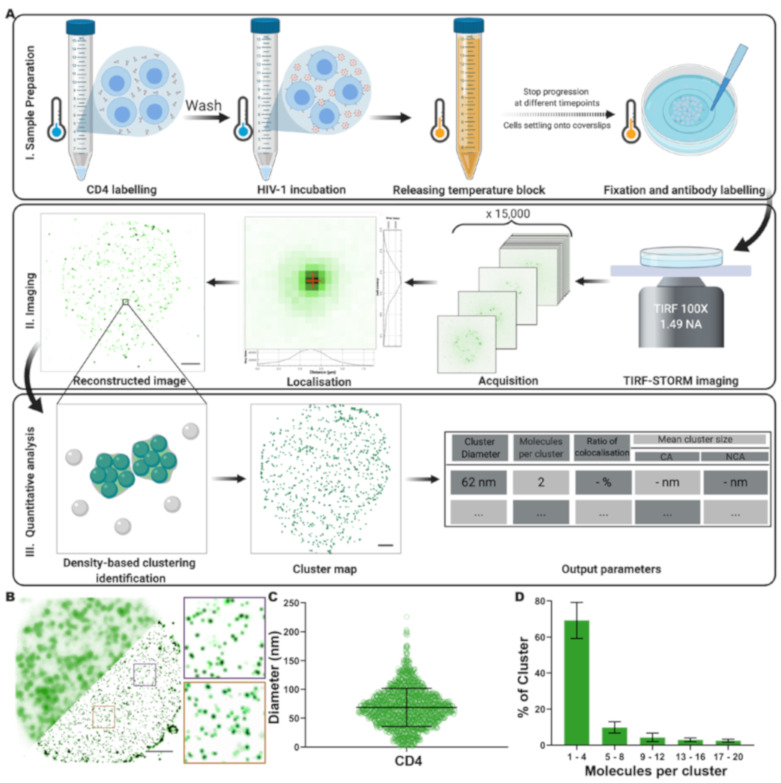
Experimental and analytical pipeline for the quantitative analysis of plasma membrane CD4. (**A**) Schematics of the experimental and analytical pipelines used. (I) SupT1-R5 cells were pre-incubated at 4 °C with OKT4 with or without HIV or gp120, prior to a brief release of the temperature block and rapid cooling prior to fixation. (II) Cells deposited on imaging surfaces were imaged using a standard TIRF-STORM acquisition, and (III) localisations were used for density-based cluster analysis and molecular counting. (**B**) Representative diffraction-limited TIRF (top half) and TIRF-STORM (bottom half) images of CD4 in an untreated SupT1-R5 cell. Insets show enlarged TIRF-STORM images of the indicated regions. Scale bar = 2 μm. (**C**) Quantification of CD4 cluster diameters in the same cells. Each point represents 1 cluster; 15 cells were measured, giving a total of 11768 clusters. Bars represent mean ± SD. (**D**) Molecular counting of CD4 clusters. Each bar represents the percentage of clusters displaying the indicated ranges of molecules per cluster. Bars represent mean ± SD. All data represent at least three independent experiments.

**Figure 2 viruses-13-00142-f002:**
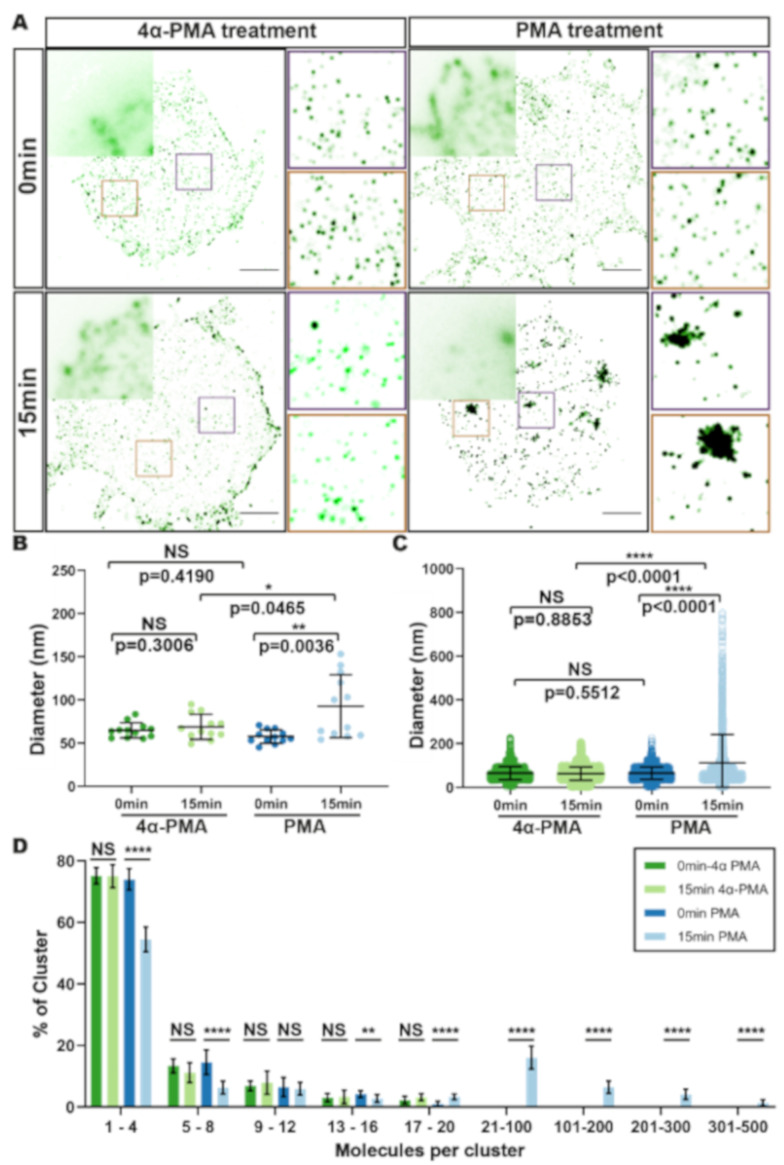
CD4 reorganisation in response to PMA and 4α-PMA. (**A**) Representative TIRF (**top left corner**) and TIRF-STORM (bottom) images of CD4 clusters in SupT1-R5 cells following treatment for 0 min (**top**) or 15 min (**bottom**) with 4α-PMA (**left**) or PMA (**right**). Insets show magnified TIRF-STORM images of the indicated regions. Scale bar = 2 μm. (**B**,**C**) Quantification of CD4 cluster diameters; (**B**) each dot represents the average cluster diameter per cell and (**C**) each dot represents the diameter of individual clusters. 15 cells were measured per condition; bars indicate the mean ± SD. (**D**) Molecule counting of CD4 clusters. Each bar represents the mean cluster fraction displaying the indicated ranges of molecules per cluster. Data are representative of at least three independent experiments. Bars represent mean ± SD. * *p* < 0.05; ** *p* < 0.01; **** *p* < 0.0001.

**Figure 3 viruses-13-00142-f003:**
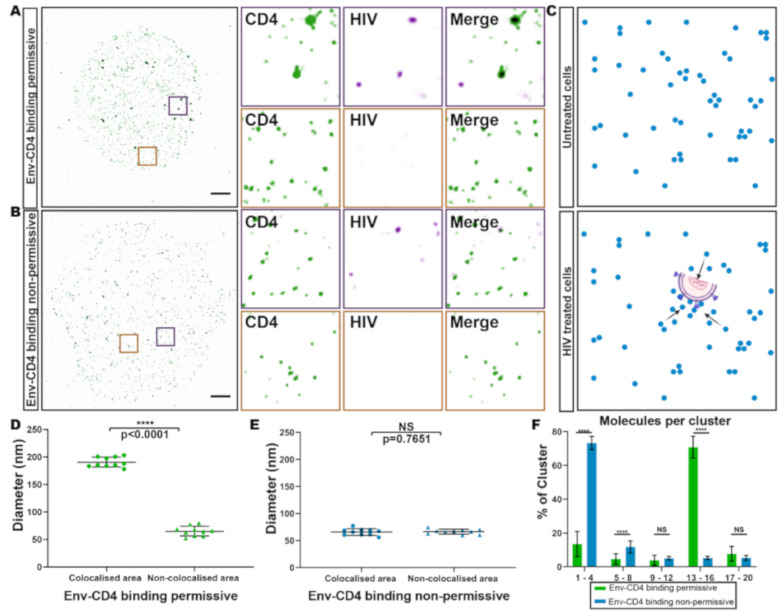
Analysis of HIV-bound CD4 clusters. (**A**,**B**) Representative TIRF-STORM images, and selected magnified regions (insets) of cell-surface CD4 (green) and HIV p24 (magenta). Scale bar = 2 μm. (**C**) Schematic of cell-surface CD4 molecules (blue) in untreated cells (**upper panel**) and HIV treated (or bound) cells (**lower panel**). (**D**,**E**) Quantification of CD4 cluster diameters in the absence (**D**) or presence (**E**) of Q4120. Each dot represents the average CD4 cluster diameter on one cell. 10 cells were measured per condition, bars indicate mean ± SD. The data are representative of at least three independent experiments. (**F**) Molecule counting of colocalised areas in the absence (Env-CD4 binding permissive) or presence of Q4120 (Env-CD4 binding non-permissive). Bars indicate mean ± SD. **** *p* < 0.0001.

**Figure 4 viruses-13-00142-f004:**
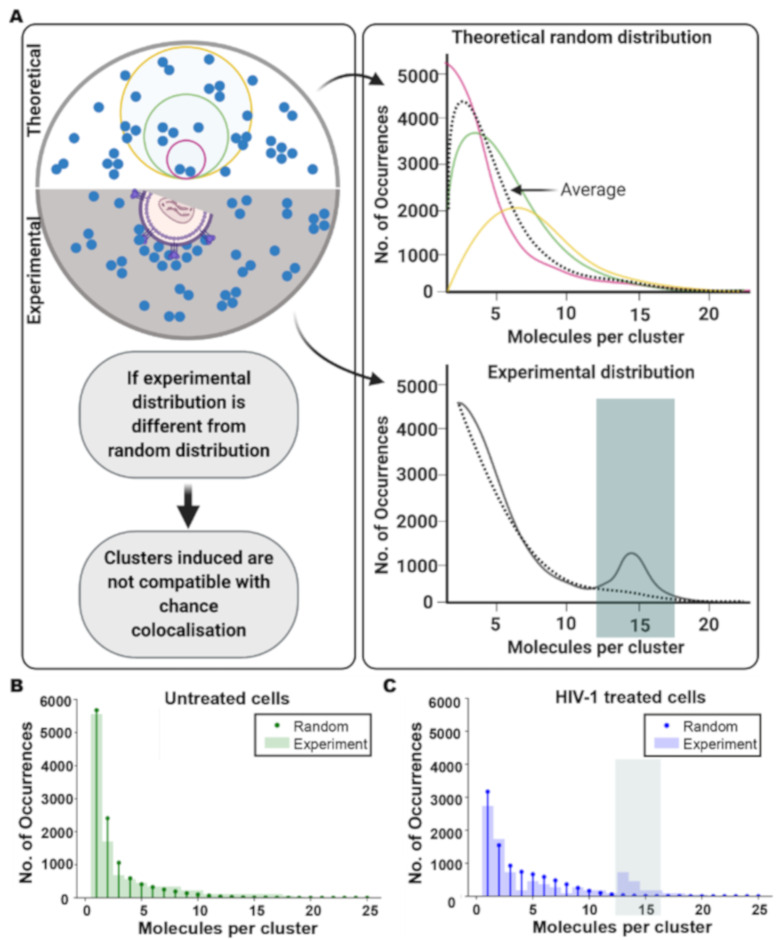
Statistical modelling of CD4 surface distribution. (**A**) Model schematic (Top): counting CD4 molecules (blue) in membrane areas of different sizes (small (pink), medium (green) or large (yellow)) compared to a theoretical model based on an averaged Poisson distribution that corresponds to a random distribution of receptors on the cell surface and to multiple measurements in membrane areas (clusters) of different sizes (the black line is the average of Poisson distributions for different cluster sizes [pink, green, yellow]). Bottom: Experimentally determined distribution: HIV binding alters the organisation of receptors, the occurrence of clusters of a certain number of molecules per cluster is altered (solid line) and the distribution differs from the expected averaged Poisson distribution (dotted line). (**B**,**C**) Comparison of modelled (‘Random’) and measured distributions of the numbers of molecules per cluster for untreated cells (**B**) and HIV-treated (bound) cells (**C**). The discrepancy between the predicted model and the experimental data in HIV-treated cells is highlighted in green shading in **A** and **C**. Data from at least three separate experiments and total 15 different cells in each condition, respectively.

**Figure 5 viruses-13-00142-f005:**
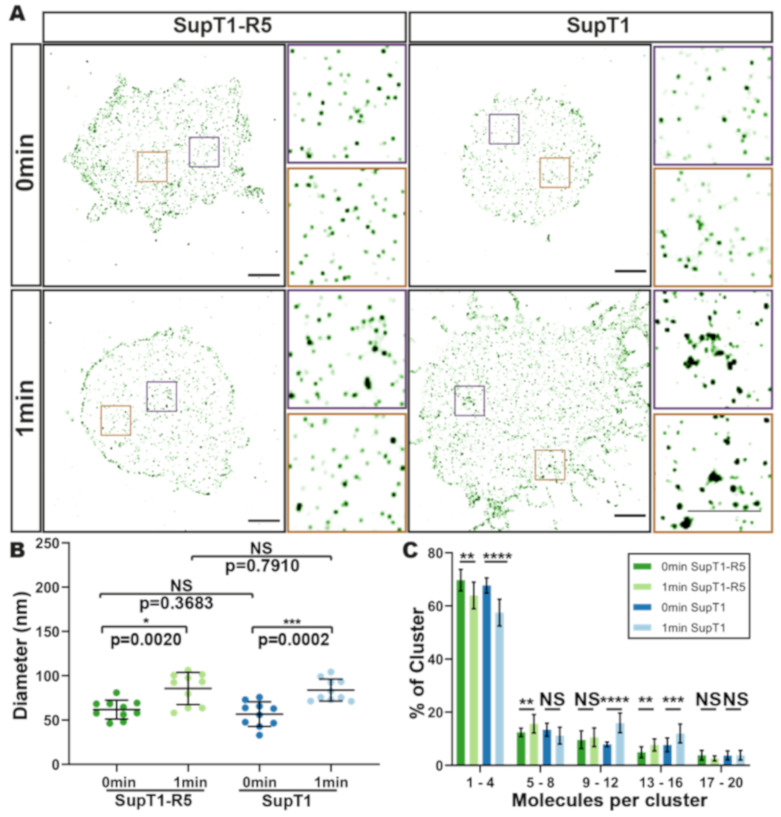
CD4 nanoscale organisation following binding of gp120 in SupT1-R5 and SupT1 cells. (**A**) Representative TIRF-STORM images of CD4 on SupT1-R5 cells (**left**) and SupT1 cells (**right**), following gp120 and OKT4-Alexa Fluor 647 pre-binding and warming for 1 min prior to fixation. Control samples were fixed directly following gp120 treatment. Insets show magnified TIRF-STORM images of the indicated regions. Scale bars = 2 μm. (**B**) Mean receptor cluster diameters per cell are plotted for each gp120-treated SupT1 and SupT1-R5 condition. Each point represents the mean value for a single cell. (**C**) Molecule counting; the cluster fraction per range of molecule numbers is plotted for each condition. The error bars plot the overall mean and SD. * *p* < 0.05; ** *p* < 0.01; *** *p* < 0.001; **** *p* < 0.0001.

## Data Availability

Data available on request due to restrictions.

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
