# Peer review of "Single-Molecule Super-Resolution Imaging of T-Cell Plasma Membrane CD4 Redistribution upon HIV-1 Binding"

_viruses, 2021, doi:10.3390/v13010142_

Round 1
Reviewer 1 Report
The manuscript by Yuan et al., reports on a nanoscale analysis of HIV-induced spatial changes in cell-surface CD4 distribution during the first initial steps of HIV infection.
The results are sound and the paper is clear and well written.
Limitations, such as for example the underestimation of the number of CD4 molecules and the lower affinity of the anti-CD4 antibody used in the study are well explained.
The methodology is strong and the study’s result that HIV induced clusters of 13-16 CD4 molecules, roughly 1:1 CD4 per Env trimer is an important piece of information in understanding how receptor molecules are recruited in cell surface-bound HIV.
Can authors justify the use of HIVJR-CSF?
Although, the data presented support the conclusions of the study, it would be nice if the authors showed similar results with a different virus. Did authors test HIVNL4-3 or any X4 strain isolates?
Can authors present data on one X4 and one R5-strain of HIV? It will strengthen even more their study.
The results that local recruitment of CD4 following gp120 is not dependent on the presence of a coreceptor, in particular CCR5 here, is interesting. Did authors look at CCR5 clusters themselves following HIV infection? Can authors speculate based on their study and what is known in the literature? Do authors expect to be the same for CCR5 and CXCR4?
A couple of technical comments.
Why did authors use 30 infectious particles per cell? Did authors test fewer or more than 30?
Cells were first pre-incubated with the anti-CD4 OKTA-AF647 antibody and then HIV was added. I assumed that HIV and the anti-CD4 antibody recognize different CD4 epitopes. Why was the anti-CD4 added first?
Author Response
We thank the reviewer for their feedback. Please see the attached file for a detailed reply.

Reviewer 2 Report
In the manuscript by Yuan et al. the authors applied super-resolution microscopy to quantify CD4 cell surface distribution and redistribution upon PMA treatment and HIV binding. They found interesting features of CD4 molecule behavior at a nanoscale level. Particularly, PMA induced a coalescence of CD4 into big clusters of different size, while HIV particle binding resulted in a formation of clusters consisting predominantly of 13-16 CD4 molecules which exceeded the size of viral particles. At the surface of untreated cells CD4 was present by clusters of 1-4 molecules. To achieve this, authors developed a pipeline protocol for cell labeling and treatment in suspension, rapid temperature shift followed by immediate fixation. Appropriate controls that included temperature, activators, and blocking antibody were used to verify the obtained effects. In addition, mathematical normal and experimental distribution values were calculated to confirm the HIV-binding effect on CD4 coalescence. The study adds some novelty in our knowledge of CD4-HIV binding early events at nanoscale level, and can be recommended for publication. However, some important technical questions need to be addressed first.
- Given that a monoclonal IgG antibody is bivalent, and an antigen carries usually one epitope recognized by a mAb, it is very likely that anti-CD4 OKT4 mAb will dimerize CD4 even on ice condition. Since in all experiments CD4 staining was performed before cell fixation and based on results in Fig.1 showing that average number of CD4 molecules per cluster is 2, would it be possible that in non-stimulated cells CD4 is distributed even more diffusely (close to individual molecules)? Was the mAb labeling of CD4 ever compared to CD4 tagged with monomeric GFP, for example, mClover? Similarly, was the CD4 fixed prior to labeling with anti-CD4 mAb?
- In experiment utilizing double-staining of CD4 and HIV p24 (Fig.3A-B) the first one was done using mouse mAb directly conjugated to Alexa dye, the second one was performed with primary mouse mAb and secondary anti-mouse fluorophore-conjugated Ab. However, CD4 stained with mouse anti-CD4-Alexa primary mAb will likely be labeled by the secondary goat anti-mouse Ab as well. Did you see such a cross-staining of these samples? If not, what is the explanation for this?
- When SupT1 cells were challenged with HIV, was the number of attached viral particles on cells pretreated with Q4120 mAb lower than the amount of HIV particles on cells labeled with not neutralizing OKT4? And how can it be explained?
- The protocol of sample preparation for microscopy should be described more clearly. For example, from either Methods or Result section it was unclear how rapid fixation has been performed.
For example, 2.7 “Treated samples were transferred to 37°C RPMI-1640 containing 5% FBS for 1 minute, before returning to cold RPMI-1640 for 5 minutes. The cells were then fixed with cold 4% PFA for 10 minutes, before warming to 37°C over 20 minutes”
How samples were transferred to 37C? Was it done in suspension? Centrifugation? Removing previous medium and adding warm medium? “Returning to cold RPMI” means removing, centrifuging and adding? If the medium contains 5% FBS, the last one will interfere with fixative reagent. Was a volume excess of fixative used to overcome serum inhibition or washing with serum-free medium was used?
Author Response

(The authors gave the same response as above.)

Reviewer 3 Report
Summary
Yuan et al. analyze CD4 receptor clustering by human immunodeficiency virus (HIV). They establish a protocol to pre-label CD4 with fluorescently conjugated antibodies and they assess short term virus-cell interactions and analyze the resulting CD4 clustering by super resolution localization imaging STORM. Based on the nanometer resolution of the CD4 receptors and based on a calibration of the system with fluorescent antibodies, they are able to estimate the number of CD4 receptors per cluster. In resting cells most CD4 spots contain less than four receptors. Next, they artificially induce CD4 clustering by activating clathrin mediated endocytosis with PMA. Clusters of up to a couple hundred CD4 receptors were observed. For HIV, they bind the virus for 1h in the cold to avoid fusion and entry events. They observe small clusters of 12-16 CD4 molecules per cluster and the clusters are around 200nm ins size. This cluster size and receptor density was reproduced with computational modeling. Together, this work presents a detailed analysis of a very early event of HIV interaction with host cells. It is a critical step towards a more quantitative understanding of virus -receptor interactions. While is important and the setup was carefully validated, some questions regarding cluster and virus sizes remain. There is some heterogeneity in the size of the virus signal and it is not clear why some bound HIV particles lack CD4 signals.
Major
In figure 3a, two CD4 clusters at HIV particle extend well beyond the particle size. How do the authors explain this expanded clusters? The HIV diameter is around 100nm while the cluster size averages at 200nm in panel 3d. How is this explained? Moreover, there is a third particle that apparently doesn’t have an associated CD4 signal. It appears that virions without CD4 were not included in the histogram in 3f. There is also a significant heterogeneity in HIV particle size and intensity. Are these considered individual particles or clusters of particles that bind cells? Was the virus preparation analyzed by electron microscopy to exclude the possibility of clusters or small aggregates? In figure S3 HIV is bound to cells in the cold and the authors show that there is no CD4 clustering at the virus location occurring. Some of the virus spots in figure S3 are not showing CD4 signal at all. Panel S4B is suggesting that close to 20% of virions lack CD4 signal. In the absence of CD4, how is the virus thought to be bound to the cells? CD4 is established to be the primary attachment receptor for HIV with co-receptors CCR5 and CXCR4 and maybe additional receptors facilitating the attachment. Finally, they use purified gp120 protein on cells with or without CCR5 co-receptors and find that the CD4 cluster formation was independent of CCR5.
Minor
Line 242: the sentence contains an extra word, it should be either but or while.
Statistical analysis of the significance in figure S5 is missing.
The data that was used to calibrate the molecular counting is critical for the analysis of the cluster size and should be presented in the supplemental document.
The localization imaging yields maps of XY-positions that are then converted to signal intensities. For illustration, one example of a raw point map image for CD4 and HIV should be provided side-by-side with the final image.
The actual optical resolution that was obtained with this microscopy and analysis setup should be indicated.
Author Response

(The authors gave the same response as above.)
